# Target-Oriented User Equilibrium Considering Travel Time, Late Arrival Penalty, and Travel Cost on the Stochastic Tolled Traffic Network

Xinming Zang, Zhenqi Guo, Jingai Ma, Yongguang Zhong * and Xiangfeng Ji

Department of Management Science and Engineering, School of Business, Qingdao University, Qingdao 266071, China; zangxinming@qdu.edu.cn (X.Z.); guozhenqi95@gmail.com (Z.G.); majingai96@gmail.com (J.M.); jixiangfeng@qdu.edu.cn (X.J.)
* Correspondence: zhongyongguang@qdu.edu.cn

**Abstract:** In this paper, we employ a target-oriented approach to analyze the multi-attribute route choice decision of travelers in the stochastic tolled traffic network, considering the influence of three attributes, which are (stochastic) travel time, (stochastic) late arrival penalty, and (deterministic) travel cost. We introduce a target-oriented multi-attribute travel utility model for this analysis, where each attribute is assigned a target by travelers, and travelers' objective is to maximize their travel utility that is determined by the achieved targets. Moreover, the interaction between targets is interpreted as complementarity relationship between them, which can further affect their travel utility. In addition, based on this travel utility model, a target-oriented multi-attribute user equilibrium model is proposed, which is formulated as a variational inequality problem and solved with the method of successive average. Target for travel time is determined via travelers' on-time arrival probability, while targets for late arrival penalty and travel cost are given exogenously. Lastly, we apply the proposed model on the Braess and Nguyen–Dupuis traffic networks, and conduct sensitivity analysis of the parameters, including these three targets and the target interaction between them. The study in this paper can provide a new perspective for travelers' multi-attribute route choice decision, which can further show some implications for the policy design.

**Keywords:** target-oriented perspective; multiple attribute; travel time; late arrival penalty; travel cost; user equilibrium; sensitivity analysis

## 1. Introduction

Traffic assignment models play a fundamental role in transportation planning and real-time applications, e.g., traffic prediction and optimal routing, and this is important for the desirable development of cities, e.g., a low-carbon city [1]. The work of [2] proposed two widely used principles for assigning vehicular flows to the traffic network, and his first principle is closely related to our study, i.e., the user equilibrium principle, which can be stated as: No one can decrease his (or her) route travel time by unilaterally changing his (or her) route choice decision. Although Wardrop's user equilibrium principle is widely used, it has some unrealistic assumptions, e.g., the traffic network is deterministic, only travel time is considered in travelers' route choice decision, travelers always grasp all information about the network conditions, and always incline to adopt the route with the shortest travel time, i.e., they are naturally assumed to be "rational people" [3]. One of the aims of this paper is to partially overcome the unrealistic assumptions of this principle.

Uncertainty is inevitable in the real-world traffic network, which could come from the demand side (e.g., peak-hour demand surge), supply side (e.g., traffic accident), or both [4,5]. Uncertainty is categorized into two kinds by [6], which are risk and ambiguity. The fundamental difference between these two kinds is whether the full distribution information of outcomes is known or not. Particularly, it is known in the risk modeling,

while unknown in the ambiguity modeling. Scholars have conducted numerous studies on travelers' route choice behavior under risk (e.g., [4,7,8]) and ambiguity (e.g., [5,9–11]), and the resultant models are fundamental to the traffic assignment, e.g., in the formulation of variational inequality (e.g., [4,7,8,12]). For example, the authors in [4] proposed the mean-excess travel time (METT) model based on the travel time budget (TTB) model in [12]. The authors in [7] assumed travelers' confidence level belongs to an interval due to day-to-day travel time variations, and proposed the subjective-utility TTB model. The work of [8] introduced a new travelers' non-expected disutility model, which can incorporate METT and TTB as the special cases. The authors in [9] formulated the shortest path problem with distributional uncertainty as a distributionally robust mixed integer linear programming. The authors in [10] discussed the robust shortest path problem with distributional uncertainty. The authors in [5] assessed the travel time reliability of the traffic network with partial information of the distribution. The authors in [11] formulated Wasserstein's distributionally robust shortest path as a mixed 0–1 convex problem. Among all the studies, risk could come from the distributions of free-flow travel time, e.g., gamma distribution and normal distribution, distribution of the link capacity degradation, e.g., uniform distribution, and the distribution of travel demand, e.g., normal distribution [4], while the ambiguity set of probability distributions could be described by the moment information, e.g., [9,10], or by the statistical distance [11]. The study in this paper focuses on the risk modeling, i.e., route choice decision making by travelers under risk.

Substantial studies have demonstrated that there are multiple attributes affecting the route choice decision made by travelers (e.g., [13–16]). For example, the authors in [13] conducted the statistical analysis to examine the route attributes which travelers consider to be important during their route choice, and the results show that three most important attributes are shorter travel time (40% of respondents choose this), travel time reliability (32% of respondents choose this), and shorter distance (31% of respondents choose this). Besides the empirical studies, scholars also make substantial progress on travelers' multi-attribute route choice behavior form the theoretical side, and two main methodologies are widely used in the current studies. One of these two is to aggregate different criteria into the generalized travel cost (e.g., [4,8,12,17,18]), and the other one is to study different criteria separately, resulting in a multi-objective route choice model (e.g., [19–21]). For example, in the TTB model proposed by the authors in [12], travel time and travel time reliability are considered, while in the METT model proposed by the authors in [4], travel time, travel time reliability, and travel time unreliability are discussed. The work of [19] treated travel time and travel time reliability separately, resulting in the bi-objective route choice model, and the authors in [21] discussed TTB and distance separately. Research in this paper also discusses travelers' multi-attribute route choice behavior, and combines all the attributes aggregately.

Another noteworthy topic closely related to this paper is to use behavioral economics theory in the study of traffic and transportation problems, e.g., prospect theory (PT) proposed in [22], and cumulative prospect theory (CPT) proposed in [23]. (C)PT can be used to explain a variety of phenomena that cannot be explained with the expected utility theory (e.g., [24]). The authors in [25] firstly proposed to incorporate CPT in the equilibrium analysis on the stochastic network, where the reference is exogenous. Following this paper, the authors in [3] proposed an endogenous reference when applying the CPT in the equilibrium analysis, and discussed a behaviorally consistent congestion pricing problem based on the equilibrium principle. In the recent decade, (C)PT are also widely used in other aspects of traffic and transportation study, e.g., choice in the unreliable transport networks [26], day-to-day learning [27], mode choice [28], and electric vehicle drivers' charging behavior of battery [29], to name a few. (C)PT and the methodology used in this paper both belong to the descriptive paradigm, i.e., they have similarities, while they also have differences following the discussion in [30], which are summarized as follows: (1) The target under this methodology serves as a specific reference under (C)PT; (2) (C)PT has two typical characteristics, namely, probability distortion function and reference-dependent

value function, while stochastic correlation among these attributes and the interaction between targets, namely the complementarity relationship between targets, are the two typical characteristics of this methodology.

The closest paper related to our study here is [31], where the author proposed a new general methodology to study travelers' multi-attribute route choice behavior on the traffic network (in Section 2.1, we review this methodology to make this paper self-contained). Based on this methodology, the author considered stochastic travel cost and travel time simultaneously in a tolled traffic network, and further discussed a target-oriented bi-attribute travel route utility model and bi-attribute user equilibrium based on aforementioned utility model. Nevertheless, although the authors in [31] proposed a general methodology for the multi-attribute route choice modeling, they only discuss a bi-attribute application, which motivates the multi-attribute study in this paper. To be specific, we apply the methodology proposed in [31] in the multi-attribute analysis, and extend the application scope of this methodology.

Additionally, the study in this paper focuses on travelers' a priori route choice behavior, which means that they will not change their route choice decision after this is determined. In contrast, there is another kind of research called adaptive route choice, where travelers can change their route choice behavior based on the information of a traffic network. For example, the authors in [32] firstly adopted (C)PT utility function to capture the impact of risk attitude and online information on route decision adjustments. The authors in [33] formulated information location models for minimizing travel cost of travelers, which can be classified into three categories according to congestion effect and vehicle number. The authors in [34] proposed the adaptive routing policy problem and its necessary conditions for optimality in the stochastic network, considering three types of partial online information, namely, delayed global, global pre-trip, and up-to-date radio. The author in [35] developed a label-setting-based algorithm to identify the adaptive least-expected time hyperpaths. The work of [36] proposed user equilibrium with recourse (UER) models in the stochastic network, where the link states is defined by probability mass function. Furthermore, the work of [37] formulated the continuous network design problem under UER as bi-level programming. In the studies of experimental economics, the authors in [38] compared travelers' route choice decision under no online information (expected user equilibrium) and perfect online information (UER), and the results demonstrate that the provision of online information brings heavy travel costs. The authors in [39] concluded the provision of information can reduce travelers' risk aversion behavior, and further reduce their valuation on information and reliability.

With all the above discussions, we adopt the target-oriented methodology to analyze the multi-attribute route choice decision of travelers in the scenario of tolled traffic network, considering three attributes, which are (stochastic) travel time, (stochastic) late arrival penalty (LAP), and (deterministic) travel cost, following the work of [18] (in the work, the authors used travel distance, which can also be reinterpreted as travel cost following our definitions, as our paper assumes travel cost is given in advance). That is, we extend the application scope of the general methodology in [31]. LAP can be used to model travel time unreliability, which brings the benefit for the practical use of our model. Moreover, travel cost denotes the congestion tolls on the traffic network. We introduce a target-oriented multi-attribute travel utility model for this analysis, where each attribute is assigned a target by travelers and travelers' objective is to maximize their travel utility that is determined by the achieved targets. Moreover, the interaction between targets is interpreted as a complementarity relationship between them, which can further affect their travel utility. The interaction between targets means that travelers are willing to acquire more targets for additional utility. Particularly, we propose a rule to determine the utility values based on the essentiality of the interaction between targets.

Next, the user equilibrium model based on aforementioned travel utility model is proposed, which is formulated as a variational inequality problem and solved with method of successive average. Target for travel time is determined via travelers' on-time arrival

probability, while targets for late arrival penalty and travel cost are given exogenously. Lastly, we apply the proposed model to the Braess and Nguyen–Dupuis traffic networks, focusing on the impact of these three targets and the target interaction among them, i.e., the sensitivity analysis.

The rest of the sections and contents of the paper are shown As follows. Review of the general methodology and target-oriented multi-attribute travel utility model considering the travel time, LAP, and travel cost are presented in Section 2. Target-oriented multi-attribute user equilibrium based on this new route choice model and its solution algorithm are listed in Section 3. In Section 4, we apply the proposed model on the Braess and Nguyen–Dupuis traffic networks, and conduct sensitivity analysis of these three targets and the target interaction between them. Finally, the major conclusions and findings are shown in Section 5.

## 2. Target-Oriented Multi-Attribute Travel Utility Model

The main content of this section is to study the target-oriented route choice decision of travelers based on the methodology described in [31], where travelers need to consider travel time, LAP, and travel cost simultaneously. Next, we review this general methodology considering $N$ attributes first, although we only study the impact of three attributes.

### 2.1. Review of the Target-Oriented Methodology

**Definition 1.** Considering $N$ attributes for a specific route, denoted by $\mathbf{X} = \{X_1, X_2, \ldots, X_N\}$, which are assumed to be stochastic and correlated, a traveler can be seen as target-oriented if his (or her) utility for an outcome, denoted by $\mathbf{x} = \{x_1, x_2, \ldots, x_N\}$, is determined by the targets acquired from this outcome, where each attribute is assigned one target by travelers, denoted by $\mathbf{d} = \{d_1, d_2, \ldots, d_N\}$.

Here, we assume all the attributes are stochastic and correlated. This assumption demonstrates the generality of this methodology, and the methodology for deterministic attributes or stochastic attributes without correlation can be obtained based on our methodology. That is, this methodology can handle the deterministic attributes as shown in the following application, as the deterministic attribute is a special case of the stochastic one. We introduce an indicator function for each attribute $X_i$, $(i = 1, 2, \ldots, N)$, which is formulated as

$$I_i = \begin{cases} 1, & \text{if } x_i \leq d_i \\ 0, & \text{otherwise} \end{cases} \tag{1}$$

When $I_i = 1$, we say this target is achieved, which means that travelers are satisfied with this route with outcome $x_i$ if only this attribute is considered, while it is not achieved when $I_i = 0$. Considering there are $N$ attributes, travelers' satisfaction depends on the targets that are achieved as shown in Definition (1). We introduce a set of indices $\{i | I_i = 1\}$, denoted by $A$, i.e., we use $A$ to denote the targets which are achieved. As each $I_i$ can be 0 or 1, there are $2^N$ different $A$. For example, when all the $I_i$ are 0, $A$ is an empty set; when $x_1 \leq d_1, x_3 \leq d_3$, and $x_i > d_i (i = 2, 4, 5, \ldots, N)$, $A$ is $\{1, 3\}$.

Accordingly, the occurrence probability for each $A$ is written as $P(x_i \leq d_i, i \in A; x_j > d_j, j \notin A)$ considering that all the outcomes are stochastic. The utility when different targets are achieved, termed as target-oriented utility, is denoted as $\xi(I_i = 1, i \in A; I_j = 0, j \notin A)$, which is rewritten as $\xi(A)$ or $\xi_{i \in A}$ for brevity hereinafter, depending on the context. Moreover, we assume that the value of $\xi(A)$ is larger when more targets are achieved, and $\xi(A) = 1$ when all the targets are achieved and $\xi(A) = 0$ when no target is achieved. Therefore, only $2^N - 1$ situations need to be considered, and with the consideration that they are mutually exclusive, the target-oriented route utility $\omega$ is formu-

lated as the sum of $2^N - 1$ values, where each value is the product of the target-oriented utility and the corresponding occurrence probability, i.e.,

$$\omega = \sum_{A \subseteq N, A \neq \varnothing} \xi(A) P(A) \tag{2}$$

If the joint probability $P$ is known, all the occurrence probabilities used in Equation (2) can be evaluated. However, it is rarely known in practice, as it takes a huge cost to grasp perfect information. We adhere to the inclusion-exclusion rule proposed in [31] and reformulate Equation (2) as

$$\omega = \sum_{A \subseteq N, A \neq \varnothing} \omega_A P(X_i \leq d_i, i \in A) \tag{3}$$

where $\omega_A = \sum_{B \subseteq A} (-1)^{|A|-|B|} \xi(B)$.

This reformulation brings two benefits. Firstly, the marginal distributions, which might be estimated with the data-driven methods, can be used to evaluate the joint probabilities. The second benefit is that $\omega_A$ can be employed to grasp the interaction between targets, which can also be interpreted as the complementarity relationship between them. Both of these benefits will be shown in detail in the next subsections.

### 2.2. Network Representation and Attributes

We use $G(V, A)$ to denote a connected and directed traffic network, where $V$ denotes the node set and $A$ denotes the link set. Let $R$ and $S$ denote the set of origins and destinations, respectively, and $rs$ denote $OD$ pair from an origin $r \in R$ to a destination $s \in S$. Let $q^{rs}$ denote the travel demand between this $OD$ pair. We use $p^{rs}$ to denote all the routes between the $OD$ pair, and the route flow vector $\left(\ldots, f_p^{rs}, \ldots\right)$ is denoted by $f$, where $f_p^{rs}$ denotes the flow on route $p \in P^{rs}$. The link travel time is stochastic as we discuss the stochastic traffic network, which is denoted by $T_a, \forall a \in A$, and the link flow is denoted by $v_a, \forall a \in A$. We use $\Delta = \left[\delta_{pa}^{rs}\right]$ to denote the link-route incidence matrix, where $\delta_{pa}^{rs} = 1$ if link $a$ is on the route $p$, and 0, otherwise. Finally, we use $c_a$ to denote the toll on link $a$, $\forall a \in A$.

As aforementioned, we study traveler's target-oriented route choice decision considering (stochastic) travel time, (stochastic) LAP, and (deterministic) travel cost. In this section, we discuss the details on these three attributes, e.g., the probability distribution functions and the corresponding targets we used.

#### 2.2.1. Stochastic Route Travel Time and Its Corresponding Target

Several studies have discussed how to derive the probability distribution for route travel time (e.g., [40,41]), and we adopt the method in [40]. The link travel time throughout this paper is the BPR performance function, which is written as

$$t_a(x_a) = t_a^0 \left[1 + \beta \left(\frac{v_a}{C_a}\right)^n\right] \tag{4}$$

where for each link, $a$, $t_a^0$ is the free-flow travel time, $C_a$ is the capacity, $v_a$ is the flow as shown before, and $t_a(x_a)$ is the travel time with flow $v_a$. $\beta$ and $n$ are deterministic parameters. The value of $\beta$ is 0.15 and the value of $n$ is 4. In [40], stochasticity comes from the link capacity degradation, and route travel time is shown to follow the normal distribution $N\left(E\left(T_p^{rs}\right), \sigma\left(T_p^{rs}\right)\right)$, where $E\left(T_p^{rs}\right)$ denotes the mean of the stochastic travel time $T_p^{rs}$ for route $p$ between $OD$ pair $rs$, and $\sigma\left(T_p^{rs}\right)$ denotes its standard deviation. The detailed derivation is relegated into Appendix A.

Next, we denote the route travel time as $\gamma_t$ and show how to determine the target for it, which is motivated by the method proposed in [3]. Travelers are assumed to own the same on-time arrival probability, denoted by $\theta (\theta \geq 0.5)$. $\theta = 0.5$ means that travelers are risk-neutral, and larger values of $\theta$ means that travelers are more risk-averse. Target for travel time is represented as the travel time budget specified by the travelers to ensure their desired on-time arrival probability. Therefore, when travelers choose a route $p$ from multiple routes of *OD* pair $rs$, target on this route, denoted by $\gamma_p^{rs}$, is formulated as

$$\gamma_p^{rs} = \min\left\{ \gamma \mid \Pr\left(T_p^{rs} \leq \gamma\right) \geq \theta \right\} \tag{5}$$

As *OD* pair $rs$ is connected by multiple routes, target for travel time of a route belonging to this *OD* pair is represented as the minimum value of the budget of all travel times on all routes, i.e., $\gamma_t^{rs} = \min\limits_{p \in P^{rs}} \left\{ \gamma_p^{rs} \right\}$. When we employ this method to obtain the target for travel time of a route, it seems that travelers specify the exogenous on-time arrival probability first, and then the target is endogenously determined with this exogenously given on-time arrival probability.

### 2.2.2. Stochastic Route LAP and Its Corresponding Target

Given the target for travel time of a route, denoted by $\gamma_t$, following the idea in [18], route LAP is defined as $L_p^{rs} = \max\left\{ 0, T_p^{rs} - \gamma_t \right\}$, $\forall p \in P^{rs}$, $r \in R$, and $s \in S$. In the work of [18], route LAP is defined, given a longest possible route travel time. While we assume this longest possible route travel time is the target for route travel time. One can also define the longest possible route travel time as the sum of target for route travel time and some buffer time, and our arguments can be amplified accordingly.

In this paper, target for LAP is assumed to be exogenously given, denoted by $\gamma_l$, which can be interpreted as the allowable delay by the company. Next, we construct the probability distribution function for route LAP. We see that the range of route LAP can be written as $[0, L_{\max}]$, where $L_{\max}$ is the maximum value of route LAP. Given the target for travel time of a route (denoted as $\gamma_t$), and the above-derived probability distribution function of route travel time, we construct the following probability distribution function for route LAP.

$$P\left(L_p^{rs} \leq \gamma_l\right) = \begin{cases} 0 & \text{if } \gamma_l < 0 \\ F_T(\gamma_t + \gamma_l) & \text{if } \gamma_l \geq 0 \end{cases} \tag{6}$$

where $F_T$ denotes the cumulative distribution function of route travel time, and $\gamma_l$ denotes the target for route LAP. Equation (6) can be verified as follows: when $\gamma_l < 0$, $P\left(L_p^{rs} \leq \gamma_l\right) = 0$ can be obtained, as 0 is minimum value of the range of route LAP; when $\gamma_l \geq 0$, we see that $P\left(L_p^{rs} \leq \gamma_l\right) = P\left(T_p^{rs} - \gamma_t \leq \gamma_l\right) = P\left(T_p^{rs} \leq \gamma_l + \gamma_t\right) = F_T(\gamma_l + \gamma_t)$. Especially when $\gamma_l = 0$, $P\left(L_p^{rs} = \gamma_l\right) = F_T(\gamma_t)$.

### 2.2.3. Deterministic Route Travel Cost and Its Corresponding Target

The route travel cost $C_p^{rs}$, $\forall p \in P^{rs}$, $r \in R$, and $s \in S$ can be obtained according to the link travel; cost, and is formulated as $C_p^{rs} = \sum\limits_{a \in A} \delta_{pa}^{rs} c_a$. In this paper, target for travel cost is also assumed to be exogenously given, denoted by $\gamma_c$, which can be interpreted as the acceptable expense by the travelers. Therefore, the probability that the target for travel cost can be reached on some particular routes is 1, while it cannot be achieved on other routes for the *OD* pair $rs$.

Considering the attributes and the corresponding targets, Equation (3) can be rewritten as

$$
\begin{aligned}
\omega_p^{rs} = {}& \xi_1 P\left(T_p^{rs} \leq \gamma_t\right) + \xi_2 P\left(L_p^{rs} \leq \gamma_l\right) + \xi_3 P\left(C_p^{rs} \leq \gamma_c\right) + (\xi_{13} - \xi_1 - \xi_3)P\left(T_p^{rs} \leq \gamma_t, C_p^{rs} \leq \gamma_c\right) + \\
& (\xi_{23} - \xi_2 - \xi_3)P\left(L_p^{rs} \leq \gamma_l, C_p^{rs} \leq \gamma_c\right) + (\xi_{12} - \xi_1 - \xi_2)P\left(T_p^{rs} \leq \gamma_t, L_p^{rs} \leq \gamma_l\right) + \\
& (1 - \xi_{13} - \xi_{23} - \xi_{12} + \xi_1 + \xi_2 + \xi_3)P\left(T_p^{rs} \leq \gamma_t, L_p^{rs} \leq \gamma_l, C_p^{rs} \leq \gamma_c\right)
\end{aligned}
\tag{7}
$$

where $\omega_p^{rs}$ denotes the target-oriented multi-attribute travel utility (ToMaTU) on route $p$ from an origin $r \in R$ to a destination $s \in S$. That is, Equation (7) is the specific form of Equation (3) considering the attributes and targets used in this study. The target-oriented utility $\xi_1$ represents the situation where only the first target is achieved, the target-oriented utility $\xi_{13}$ represents the situation where the first and the third targets are achieved as discussed before, and others have the similar meanings. Moreover, hereinafter $P\left(T_p^{rs} \leq \gamma_t\right)$ and $P\left(L_p^{rs} \leq \gamma_l\right)$ are called target achievement probability for travel time (TAPt), and LAP (TAPl), respectively.

### 2.3. Joint Probability Evaluation Derived from the Marginal Distributions

In this section, we show how to evaluate the joint probabilities in Equation (7) derived from the marginal distributions, namely the marginal distribution of LAP, travel cost, and travel time mentioned in last section.

First, we discuss the effect of the marginal distribution of route travel cost. We know that route travel cost is deterministic, and thus, determined by the scenario of whether the target of travel cost is finally achieved, i.e., $P\left(C_p^{rs} \leq \gamma_c\right)$ is 0 or 1, we have the following two situations. When target for travel cost is achieved, ToMaTU considering LAP, travel cost, and travel time is rewritten as

$$
\begin{aligned}
\omega_p^{rs} = {}& \xi_1 P\left(T_p^{rs} \leq \gamma_t\right) + \xi_2 P\left(L_p^{rs} \leq \gamma_l\right) + \xi_3 + (\xi_{13} - \xi_1 - \xi_3)P\left(T_p^{rs} \leq \gamma_t\right) + (\xi_{23} - \xi_2 - \xi_3)P\left(L_p^{rs} \leq \gamma_l\right) + \\
& (\xi_{12} - \xi_1 - \xi_2)P\left(T_p^{rs} \leq \gamma_t, L_p^{rs} \leq \gamma_l\right) + (1 - \xi_{13} - \xi_{23} - \xi_{12} + \xi_1 + \xi_2 + \xi_3)P\left(T_p^{rs} \leq \gamma_t, L_p^{rs} \leq \gamma_l\right) \\
= {}& (\xi_{13} - \xi_3)P\left(T_p^{rs} \leq \gamma_t\right) + (\xi_{23} - \xi_3)P\left(L_p^{rs} \leq \gamma_l\right) + \xi_3 + (1 - \xi_{13} - \xi_{23} + \xi_3)P\left(T_p^{rs} \leq \gamma_t, L_p^{rs} \leq \gamma_l\right)
\end{aligned}
\tag{8}
$$

When the target for travel cost fails to be achieved (denoted with tilde symbol), ToMaTU considering LAP, travel cost, and travel time is rewritten as

$$
\widetilde{\omega}_p^{rs} = \xi_1 P\left(\widetilde{T}_p^{rs} \leq \gamma_t\right) + \xi_2 P\left(\widetilde{L}_p^{rs} \leq \gamma_l\right) + (\xi_{12} - \xi_1 - \xi_2)P\left(\widetilde{T}_p^{rs} \leq \gamma_t, \widetilde{L}_p^{rs} \leq \gamma_l\right)
\tag{9}
$$

From Equation (8) and Equation (9), we see the effect of marginal distribution of route travel cost on the ToMaTU, and we also see that joint probability $P\left(T_p^{rs} \leq \gamma_t, L_p^{rs} \leq \gamma_l\right)$ needs to be evaluated further. For this probability, we follow the method in [42], and obtain that the random vector composed of route travel time and LAP is comonotonic. Thus, the joint probability $P\left(T_p^{rs} \leq \gamma_t, L_p^{rs} \leq \gamma_l\right)$ equals to $P\left(T_p^{rs} \leq \gamma_t\right)$.

Therefore, we can further simplify the ToMaTU, i.e., Equations (8) and (9) in our application. In the situation where the target for travel cost is successfully achieved, ToMaTU considering LAP, travel cost, and travel time is further rewritten as

$$
\begin{aligned}
\omega_p^{rs} = {}& (\xi_{13} - \xi_3)P\left(T_p^{rs} \leq \gamma_t\right) + (\xi_{23} - \xi_3)P\left(L_p^{rs} \leq \gamma_l\right) + \xi_3 + (1 - \xi_{13} - \xi_{23} + \xi_3)P\left(T_p^{rs} \leq \gamma_t, L_p^{rs} \leq \gamma_l\right) \\
= {}& (\xi_{13} - \xi_3)P\left(T_p^{rs} \leq \gamma_t\right) + (\xi_{23} - \xi_3)P\left(L_p^{rs} \leq \gamma_l\right) + \xi_3 + (1 - \xi_{13} - \xi_{23} + \xi_3)P\left(T_p^{rs} \leq \gamma_t\right) \\
= {}& (1 - \xi_{23})P\left(T_p^{rs} \leq \gamma_t\right) + (\xi_{23} - \xi_3)P\left(L_p^{rs} \leq \gamma_l\right) + \xi_3
\end{aligned}
\tag{10}
$$

In the situation where the target for travel cost fails to be achieved (denoted with tilde symbol), ToMaTU considering LAP, travel cost, and travel time is further rewritten as

$$
\begin{aligned}
\widetilde{\omega}_p^{rs} &= \xi_1 P\left(\widetilde{T}_p^{rs} \le \gamma_t\right) + \xi_2 P\left(\widetilde{L}_p^{rs} \le \gamma_l\right) + (\xi_{12} - \xi_1 - \xi_2)P\left(\widetilde{T}_p^{rs} \le \gamma_t, \widetilde{L}_p^{rs} \le \gamma_l\right) \\
&= \xi_1 P\left(\widetilde{T}_p^{rs} \le \gamma_t\right) + \xi_2 P\left(\widetilde{L}_p^{rs} \le \gamma_l\right) + (\xi_{12} - \xi_1 - \xi_2)P\left(\widetilde{T}_p^{rs} \le \gamma_t\right) \\
&= (\xi_{12} - \xi_2)P\left(\widetilde{T}_p^{rs} \le \gamma_t\right) + \xi_2 P\left(\widetilde{L}_p^{rs} \le \gamma_l\right)
\end{aligned}
\tag{11}
$$

From Equations (10) and (11), we see the hybrid effect of marginal distribution of route LAP and marginal distribution of route travel time on the ToMaRU. After this, we see that all the probabilities in ToMaTU can be obtained.

### 2.4. Target Interaction

In this paper, we follow the idea in [31] to model the interaction between targets, which can also be interpreted as the complementarity relationship between them. One of the essential meanings of target interaction is that achieving more targets could result in receiving more utility to the travelers, which is further reflected in the supermodularity of aforementioned utility function.

If, for all given subset $A$ an $B$ of $N$, the utility function always satisfies the inequation $\xi(A) + \xi(B) \le \xi(A \cup B) + \xi(A \cap B)$, then target interaction is called complementarity in the multi-attribute modeling. Here, we eliminate all the identical equations, e.g., $B = \{1\}$ and $A = \{1,2,3\}$. The complementarity relationship is motivated by the idea in economics, which means that combining two or more goods can bring more values, even though they have less or no value by themselves [43]. The complementarity relationship is further divided into two additional cases, namely, imperfect complementarity relationship and perfect complementarity relationship, respectively. In the former case, the inequation $\xi(A) + \xi(B) \le \xi(A \cup B) + \xi(A \cap B)$ is satisfied for all subsets $A$ and $B$ of $N$, while in the latter case, the inequation $\xi(A) + \xi(B) < \xi(A \cup B) + \xi(A \cap B)$ is satisfied for all subsets $A$ and $B$ of $N$. We see that the imperfect complementarity corresponds to the modularity of the utility function. Moreover, comparing the imperfect complementarity relationship with the perfect complementarity relationship, it is clear that travelers in the perfect complementarity relationship have greater willingness to achieve more targets, as achieving more targets strictly represents that travelers can receive more utility; we also know that mixed situations, namely perfect complementarity among some targets and, meanwhile, imperfect complementarity among other targets, are not discussed, e.g., $\xi_1 + \xi_2 = \xi_{12}$ and $\xi_1 + \xi_3 < \xi_{13}$. That is, if the target interaction is a perfect complementarity relationship, achieving more targets will bring more utilities to travelers strictly.

In the following, we propose a rule to determine the values of different target-oriented utilities. We assume that $\xi_1/\xi_2 = \alpha_1$, and $\xi_1/\xi_3 = \alpha_2$. Here $\alpha_i(i = 1 \text{ or } 2)$ measures the relative importance between different target achievements, e.g., larger value of $\alpha_1$ means that the target achievement of travel time on route deserves more attention, in contrast to the achievement of target set for LAP on route. Hereinafter, we term $\alpha_1$ and $\alpha_2$ as utility ratios. The relative importance between target achievement of LAP on route and that of travel cost on route can be obtained as $\alpha_2/\alpha_1$, and thus we only need two utility ratios. Starting from the imperfect complementarity relationship, we solve the subsequent system of equations to obtain different target-oriented utilities.

$$
\begin{cases}
\xi_1 + \xi_2 + \xi_3 = 1 \\
\xi_1 + \xi_2 = \xi_{12}, \ \xi_1 + \xi_3 = \xi_{13}, \ \xi_2 + \xi_3 = \xi_{23} \\
\xi_1/\xi_2 = \alpha_1, \ \xi_1/\xi_3 = \alpha_2
\end{cases}
\tag{12}
$$

While in the perfect complementarity relationship, we introduce two additional parameters, $\beta_b$ and $\beta_s$, and use prime symbol to denote the values in this relationship. We formulate $\xi_{12}'$(termed as bi-attribute target-oriented utility) as $\xi_{12}/\beta_b$, and formulate $\xi_1'$ (termed as single-attribute target-oriented utility) as $\xi_1/(\beta_b\beta_s)$ in the perfect comple-

mentarity relationship. Other target-oriented utilities in this relationship can be defined similarly. According to the definition of perfect complementarity relationship, we have $\beta_b > 1$, and $\beta_s > 1$, meanwhile, $\beta_b > 2 - (1/\beta_s)$, where 2 is equal to $\xi_{12} + \xi_{13} + \xi_{23}$, and 1 is equal to $\xi_1 + \xi_2 + \xi_3$, based on the system of equations (12). The values of $\beta_b$ and $\beta_s$, and the relationship between them, are determined according to the definition of perfect complementarity relationship. In particular, if we choose $\beta_b = 1$ and $\beta_s = 1$, the imperfect complementarity relationship is reproduced. Hereinafter, we term $\beta_b$ and $\beta_s$ as the complementarity ratios. Furthermore, $\beta_b$ is called bi-attribute complementarity ratio, as it can be used with bi-attribute target-oriented utility, while $\beta_s$ is called single-attribute complementarity ratio as it can only be used with single-attribute target-oriented utility. The bi-attribute complementarity ratio $\beta_b$ shows the interaction among three targets, while the single-attribute complementarity ratio $\beta_s$ shows the interaction between two targets. Moreover, we see that larger values of $\beta_b$ and $\beta_s$ implicates travelers have more motivation to reach these targets, as achieving more targets strictly brings extra utility. With this parameter setting rule, all the feasible results of travelers' target-oriented utilities could be achieved if we change the values of the four parameters properly.

From Equations (10) and (11), we see that it seems that some target-oriented utilities, e.g., $\xi_1$, have no impact on travelers' route choice decision. However, according to the proposed parameter setting rule, these target-oriented utilities also have their impact, not in a direct manner, but in an indirect manner, as a change their values will cause the change of other target-oriented utilities shown in Equations (10) and (11). This demonstrates the validity of our parameter setting rule, which means that this rule might be consistent with travelers' behavior.

## 3. Target-Oriented Multi-Attribute User Equilibrium

The main content in this section is to examine the long-run impact of the route choice model proposed in last section, and propose the target-oriented multi-attribute user equilibrium (ToMaUE) considering LAP, travel cost, and travel time, which can be seen as an expansion of Wardrop user equilibrium.

### 3.1. Equilibrium Condition

With the above consideration, a user equilibrium will be reached in the long run where no traveler can unilaterally change route to increase their ToMaTU, termed as ToMaUE. With the notations and definitions described in Section 2.2, ToMaUE can be stated as follows, where $\pi_{rs}$ denote the maximum ToMaTU for *OD* pair *rs*.

**Definition 2.** (ToMaUE) It is assumed that all travelers choose the route with the purpose of maximizing their ToMaTU on the traffic network, i.e., considering any *OD* pair, ToMaTUs of the routes with flows are equal and maximum, which are more than or equal to those of the routes without flows.

Under ToMaUE, the following conditions need to be satisfied:

$$\pi_{rs} - \omega_p^{rs}(f^*) = \begin{cases} = 0 \ \text{if} \ \left(f_p^{rs}\right)^* > 0 \\ \geq 0 \ \text{if} \ \left(f_p^{rs}\right)^* = 0 \end{cases} \forall s \in S, \ \forall r \in R, \ \forall p \in P^{rs} \tag{13}$$

where the equilibrium route flows are denoted by the tag *.

Furthermore, we reformulate the aforementioned ToMaUE conditions as the nonlinear complementarity problem (NCP), as shown in the following:

$$q^{rs} = \sum_{p \in P^{rs}} f_p^{rs}, \forall r \in R, s \in S \tag{14}$$

$$v_a = \sum_{r \in R} \sum_{s \in S} \sum_{p \in P^{rs}} \delta_{pa}^{rs} f_p^{rs}, \forall a \in A \tag{15}$$

$$f_p^{rs} \geq 0, \forall p \in P^{rs}, r \in R, s \in S \tag{16}$$

$$f_p^{rs}\left(\pi_{rs} - \omega_p^{rs}(f^*)\right) = 0, \forall p \in P^{rs}, r \in R, s \in S \tag{17}$$

$$\pi_{rs} - \omega_p^{rs}(f^*) \geq 0, \ \forall p \in P^{rs}, \ r \in R, \ s \in S \tag{18}$$

In the above NCP system, Equation (14) describes that for each *OD* pair *rs*, the total travel demand is equal to the sum of flows on the routes that connect this *OD* pair, i.e., the travel demand conservation. Equation (15) describes that the link flow is the sum of flows on the routes that use this link. Equation (16) describes that all the route flows are non-negative. Equations (14)–(16) show the feasible flow set of given traffic network, denoted as $\Omega$.

Therefore, a variational inequality (VI) problem $VI(f, \Omega)$ is obtained by reformulating the ToMaUE conditions, i.e., the equilibrium route flow vector $f^* \in \Omega$ needs to satisfy the inequation $\hat{\omega}(f^*)^T(f - f^*) \geq 0, \forall f \in \Omega$, where $\hat{\omega}(f^*) = -\omega(f^*)$. In addition, the properties of the VI formulation have been well studied in the literature, and one can refer to [44] for more information. Here, we discuss three properties of our VI problem, namely the equivalence between our VI problem and the ToMaUE conditions, the solution existence and solution uniqueness.

Th equivalence between our VI problem and the ToMaUE conditions (14)–(18) can be validated by examining the optimality condition of the linear programming problem $\min_{f \in \Omega} \hat{\omega}(f^*)^T f$. As aforementioned, we learn that the probability distribution function about the travel time on route is parameterized by $f$, and we assume that this function is continuous with respect to $f$. Moreover, the probability distribution function of route LAP is also continuous with respect to $f$, based on Equation (6). Therefore, we obtain that the ToMaTU is continuous with respect to $f$. Considering that $\Omega$ is compact and convex, there exists at least one solution to the above VI problem [45]. The solution uniqueness requires that the mapping in the VI problem is strictly monotone. Due to the complex formulation of the ToMaTU, $\hat{\omega}(f)$ might not be strictly monotone on $\Omega$. In particular, most user equilibrium models may not meet this requirement in the presence of uncertainty (e.g., [4,46]). Thus, to determine a reasonable equilibrium solution from the set ($\Omega$) of the feasible flow is a promising research direction, and it is also an open issue that has received sustained attention to the best of our knowledge.

### 3.2. Solution Algorithm

As shown in the obtained VI problem, the ToMaTU condition is non-additive, i.e., the VI problem follows the route-based form. In order to address it, we intend to utilize the approach of successive average (e.g., [47]), which is shown as follows. During the solution process, route enumeration is needed due to this non-additive issue, which causes high computational cost, especially for the large-scale complex network, whereas the objective of our study here is to provide insights into target-oriented route choice decision of travelers, considering these three attributes, and to design a computationally efficient algorithm is beyond our scope, which is research reserved for the future.

- Stage 0: Initialization. Set the value of the convergence tolerance as $\epsilon_0$, which means the maximum number of iterations, as well as $k = 1$.
- Stage 1: All-Or-Nothing assignment. Conduct all-or-nothing assignment in terms of the present vacant link flow $v_a^1$, as well as obtain the flow $f_p^{rs,1}$ of route connecting each *OD* pair.
- Stage 2: Link flow, route travel time distribution and route LAP distribution updating. According to the present flow $f_p^{rs,k}$ of route, orderly update the flow $v_a$ of its link, the route travel time distribution, and the route LAP distribution.

- Stage 3: Calculation for ToMaTU. Acquire the value of ToMaTU $\omega_p^{k,rs}$ based on Equation (10) or (11), and obtain the maximum route utility $\pi^{k,rs}$.
- Stage 4: Check the Convergence. Set $\epsilon = \epsilon + \dfrac{q^{rs}\pi^{k,rs} - \sum_q f_q^{rs}\omega_p^{k,rs}}{q^{rs}\pi^{k,rs}}$. Stop the algorithm if $\epsilon \leq \epsilon_0$ or the maximum number of iterations is reached, if not, go to Stage 5.
- Stage 5: Route flow updating. Detect the direction of search ($d_p^{k,rs}$) and the step size ($s^k$) judging from the present values, and the flow of route is updated as the rule $f_p^{rs,k+1} = f_p^{k,rs} + s^k d_p^{k,rs}$, and go the Stage 2. Set the direction of search as $d_p^{k,rs} = \tilde{f}_p^{k,rs} - f_p^{k,rs}$, where $\tilde{f}_p^{k,rs}$ is called as auxiliary flow. If $\omega_p^{k,rs} = \pi^{k,rs}$, $\tilde{f}_p^{k,rs} = q^{rs}/m^{rs}$ with $m^{rs}$ being the number of routes that have the maximum value of ToMaRUs in step $k$ for *OD* pair *rs*, and otherwise, $\tilde{f}_p^{k,rs}$ is 0. The step size is set as $s^k = 1/(k+1)$.

## 4. Numerical Analysis

The main content in this section is running proposed model on Braess traffic network, as well as the more general Nguyen and Dupuis's network, examining the models' performance and conducting sensitivity analysis of the parameters including these three targets and the target inter-action between them. The convergence tolerance is $10^{-6}$, and the total number of iterations is limited to $10^6$.

### 4.1. Test on the Braess Traffic Network

Firstly, we apply our proposed model to a stochastic tolled Braess network, and its performance is shown in Figure 1. There are four nodes and five links on this network, and one *OD* pair $(1,4)$ is connected by three routes. The total units of demand between this *OD* pair is 1500. Other characteristics of this network are summarized in Table 1, where $\phi$ reflects the link capacity degradation. Larger value of $\phi$ shows smaller extent of link capacity degradation. Route 1 consists of Link 1 and Link 2; Route 2 consists of Link 1, Link 3, and Link 5; and Route 3 consists of Link 4 and Link 5.

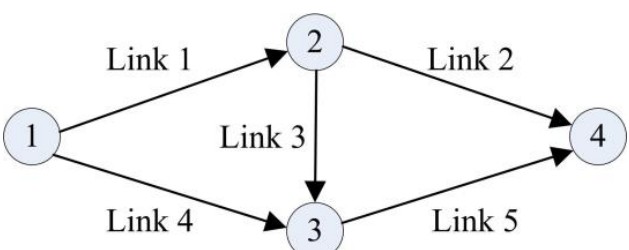

**Figure 1.** Braess traffic network.

**Table 1.** Testing data on Braess traffic network.

| Link Number | Free-Flow Travel Time | Capacity | Toll | $\phi$ |
|:---:|:---:|:---:|:---:|:---:|
| 1 | 5 | 600 | 2 | 0.8 |
| 2 | 12 | 400 | 2 | 0.7 |
| 3 | 7 | 400 | 2 | 0.9 |
| 4 | 10 | 400 | 3 | 0.7 |
| 5 | 8 | 600 | 2 | 0.8 |

We conduct the sensitivity analysis based on a benchmark, where we assume $\theta = 0.95$ (i.e., the risk-averse travelers), $\gamma_l = 5$, $\gamma_c = 5$ (Route 1 and Route 3 can successfully achieve their target for travel cost, Route 2 cannot), $\alpha_1 = 3$, $\alpha_2 = 2$, and $\beta_b = \beta_s = 1$ (i.e., in the imperfect complementarity relationship). We solve the system of Equation (12) to obtain the target-oriented utilities, and obtain $\tilde{\xi}_2 = 2/11$, $\tilde{\xi}_3 = 3/11$, $\tilde{\xi}_{12} = 8/11$, and $\tilde{\xi}_{23} = 5/11$.

### 4.1.1. Sensitivity Analysis via Changing $\gamma_c$

In this test, we compare three cases by changing the value of $\gamma_c$, which are Case 1 ($\gamma_c = 4$), Case 2 ($\gamma_c = 5$), and Case 3 ($\gamma_c = 6$), and other parameter values follow the settings in the benchmark. Target for travel cost can be achieved only on Route 1 in Case 1, can be achieved on Route 1 and 3 in Case 2, and can be achieved on all routes in Case 3. That is, we aim to show the impact of target for travel cost. The equilibrium route utility is 0.6994 in Case 1, is 0.6997 in Case 2, and is 0.97 in Case 3, and the equilibrium route flows are shown in Figure 2a. From this figure, we see that, as the value of target for travel cost increases, i.e., this target can be achieved on more and more routes, and flows will shift from the route where this target cannot be achieved to the routes where this target can be achieved. Meanwhile, the larger target value of travel cost, the higher utility value of the equilibrium route, which is also due to the target achievement on more routes. However, the extent of this increase depends on the specific value of the target $\gamma_c$, e.g., in our testing, the equilibrium route utility almost remains the same in Case 1 and Case 2.

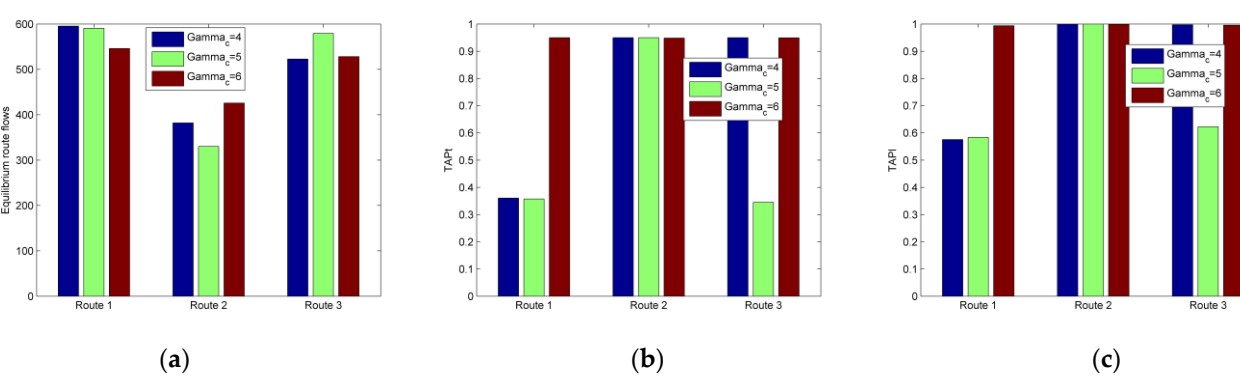

**Figure 2.** Testing results of changing $\gamma_c$: (**a**) Equilibrium route utility; (**b**) TAPt; and (**c**) TAPl.

We further discuss the changes of TAPt and TAPl by changing the value of $\gamma_c$, which is shown in Figure 2b,c, respectively. From these figures, we find that the value of TAPt is about 0.35, when the target set for travel cost can be successfully reached. That is, although travelers are risk-averse, namely the on-time arrival probability is 0.95, they will become risk-seeking to achieve three targets simultaneously. We also see that TAPl on this kind of route is also not very large, around 0.6, which means that travelers bear the risk of violating the allowable delay of the company to achieve three targets simultaneously. Meanwhile, we find that the value of TAPt and TAPl is 0.95, when the target set for travel cost cannot be reached, i.e., travelers' on-time arrival probability, and almost 1, respectively. Nevertheless, route flows on these routes are not very large, as shown in Figure 2a. By properly increasing the value of $\beta_b$ and $\beta_s$, we can also run this testing in the situation that involves the perfect complementarity relationship, and obtain the similar trends as aforementioned.

### 4.1.2. Sensitivity Analysis via Changing $\theta$

In this test, we show the impact of travelers' on-time arrival probability by changing its value from 0.5 (i.e., the risk-neutral travelers) to 0.95, and other parameter values follow the settings in the benchmark. The value of equilibrium route utility always increases as the value of $\theta$ grows, and the minimum and maximum values are 0.45 and 0.7, respectively, i.e., the emergence of risk-aversion increases travelers' equilibrium route utility. Next, we focus on the change of equilibrium route flows, which are shown in Figure 3a. From this figure, we find that the increase in $\theta$ will result in flows transforming from route 1 and 3 to route 2 initially, and then flows will shift back when the value of $\theta$ approaches 0.95. The reason for this is that a larger value of $\theta$ means larger budgeted travel time, i.e., larger target $\gamma_t$, and thus travelers can obtain certain utility, which even holds up for routes where the target set for travel cost cannot be reached. Therefore, travelers will incline to select route 2. Nevertheless, when the value of $\theta$ becomes large enough, e.g., around 0.95, TAPl on route

2 becomes 1, i.e., the target set for LAP of route is large enough, and cannot increase any more, as shown in Figure 3c, and thus more travelers will choose route 1 and route 3 where the target set for travel cost of route could be reached for a larger utility.

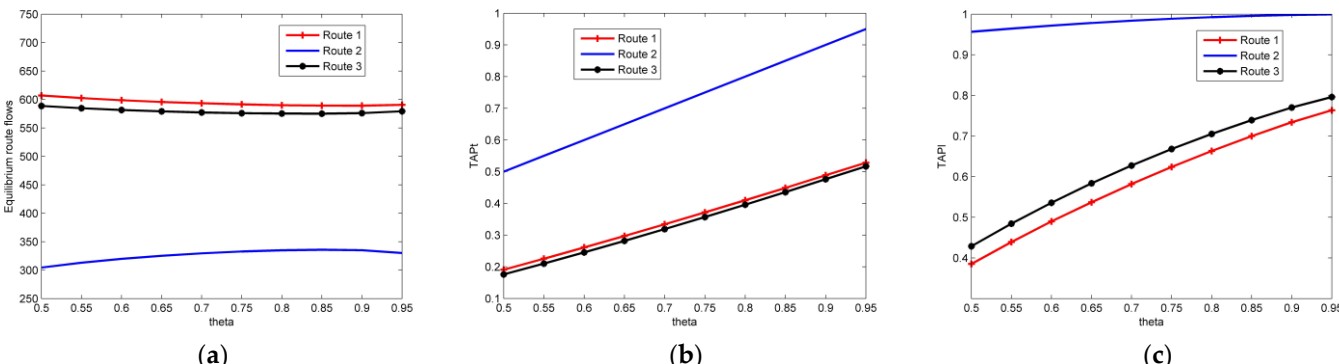

**Figure 3.** Testing results of changing $\theta$: (**a**) Equilibrium route utility; (**b**) TAPt; and (**c**) TAPl.

We further discuss the changes of TAPt and TAPl by changing the value of $\theta$, which is shown in Figure 3b,c, respectively. We can see that, with the increase in $\theta$, TAPt and TAPl both increase. For some routes, where target for travel cost could be reached successfully, TAPt is less than travelers' on-time arrival probability, and TAPl is also not very large, i.e., below 0.8, while for some routes where the target set for travel cost cannot be reached, TAPt is always travelers' on-time arrival probability, and TAPl is close to 1. That is, in order to achieve the three targets simultaneously, travelers will become less risk-averse or even risk-seeking, and will bear the risk of violating the allowable delay of the company. By properly increasing the value of $\beta_b$ and $\beta_s$, we can also run this testing in the situation that involves the perfect complementarity relationship, and obtain the similar trends as aforementioned.

### 4.1.3. Sensitivity Analysis via Changing $\gamma_l$

In this test, we show the impact of travelers' target for LAP by changing its value from 0 to 10, and other parameter values follow the settings in the benchmark. The value of equilibrium route utility keeps increasing, and the minimum and maximum values are 0.69 and 0.70, respectively, i.e., increase in the value of $\gamma_l$ brings little utility to the travelers. Next, we focus on the change of equilibrium route flows, which are shown in Figure 4a. From this figure, we find that the increase in $\gamma_l$ will result in flows transforming from route 2 to route 1 and 3. As shown in Figure 4b,c, for route 2, as the value of $\gamma_l$ increases, the value of TAPt is always 0.95, and the value of TAPl quickly converges to 1, while TAPl on route 1 and route 3 become larger and larger, i.e., route utility on route 1 and route 3 become larger and larger. Therefore, travelers will incline to select route 1 and 3, and TAPt on these routes will decrease.

We further discuss the changes of TAPt and TAPl by changing the value of $\gamma_l$, which is shown in Figure 4b,c, respectively. From these figures, we find that the value of TAPt of the routes where the target set for travel cost cannot be reached, is always 0.95, i.e., travelers' on-time arrival probability, and TAPl on this kind of route quickly becomes 1. Meanwhile, we see that on some routes where the target set for travel cost can be successfully reached, TAPl will become larger and larger as the value of $\gamma_l$ increases. In other words, to achieve the three targets simultaneously, travelers tend to transform into less risk-averse or even risk-seeking, and an increase in the value of $\gamma_l$ will decrease travelers' risk of violating the allowable delay of the company, i.e., travelers will arrive within the allowable delay of the company with a high probability. By properly increasing the value of $\beta_b$ and $\beta_s$, we can also run this testing in the situation that involves the perfect complementarity relationship, and obtain the similar trends as aforementioned.

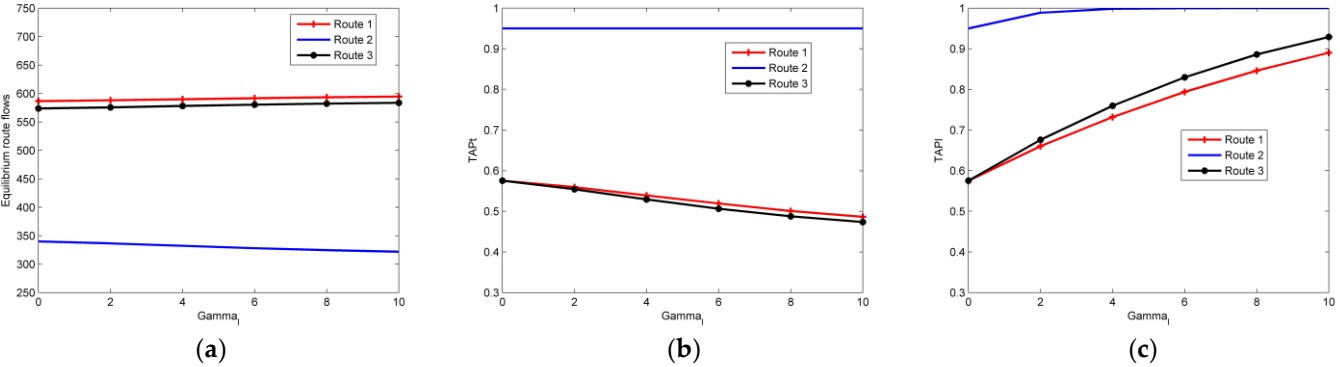

**Figure 4.** Testing results of changing $\gamma_l$: (**a**) Equilibrium route utility; (**b**) TAPt; and (**c**) TAPl.

#### 4.1.4. Sensitivity Analysis via Changing $\alpha_1$ and $\alpha_2$

In this test, we show the impact of changing the value of $\alpha_1$ and $\alpha_2$, i.e., changing the utility ratios or the relative importance between different target achievements, and other parameter values follow the settings in the benchmark. The value of $\alpha_1$ is changed from 2 to 4, and the value of $\alpha_2$ is changed from 1 to 3. Given the specific values of $\alpha_1$ and $\alpha_2$, we can calculate the value of all target-oriented utilities after addressing equation system (12).

The change of equilibrium route utility is shown in Figure 5, where its minimum value is 0.58, and its maximum value is 0.76. From this figure, we find that the value of equilibrium route utility always decreases with the value of $\alpha_1$ grows, while the value of equilibrium route utility always increases with the value of $\alpha_2$ grows. Figure 5 shows the impact of different target achievement on the equilibrium route utility. For example, recalling the definition of utility ratios, increase in the value of $\alpha_1$ shows that achieving the targets for the travel cost and travel time on route deserves more attention, in contrast to the achievement of target set for LAP on route, while increase in the value of $\alpha_2$ shows that achieving the targets for the LAP and travel time deserves more attention, in contrast to the achievement of target set for travel cost on route. We also see that the utility change is moderate as the value of $\alpha_1$ increases, and this change is relatively large as the value of $\alpha_2$ increases.

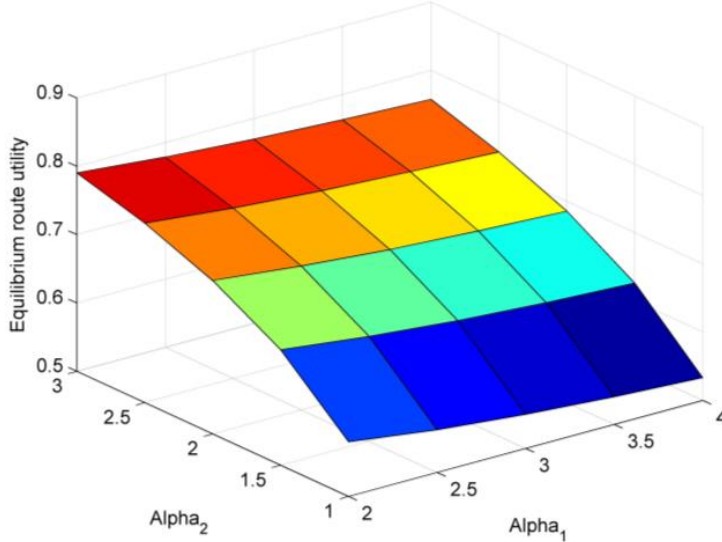

**Figure 5.** Testing results of equilibrium route utility with regard to changing $\alpha_1$ and $\alpha_2$.

The change of equilibrium route flows is presented in Figure 6a–c. From these figures, we can find that the increase in $\alpha_1$ will result in flows transforming from route 2 to route 1 and 3, as the achievement of target set for travel cost on route deserves more attention as

aforementioned, while the increase in $\alpha_2$ will result in flows transforming from route 1 and 3 to route 2, as the achievement of target set for travel cost on route becomes rarely noticed as aforementioned, recalling that the target set for travel cost fails to be achieved on route 2.

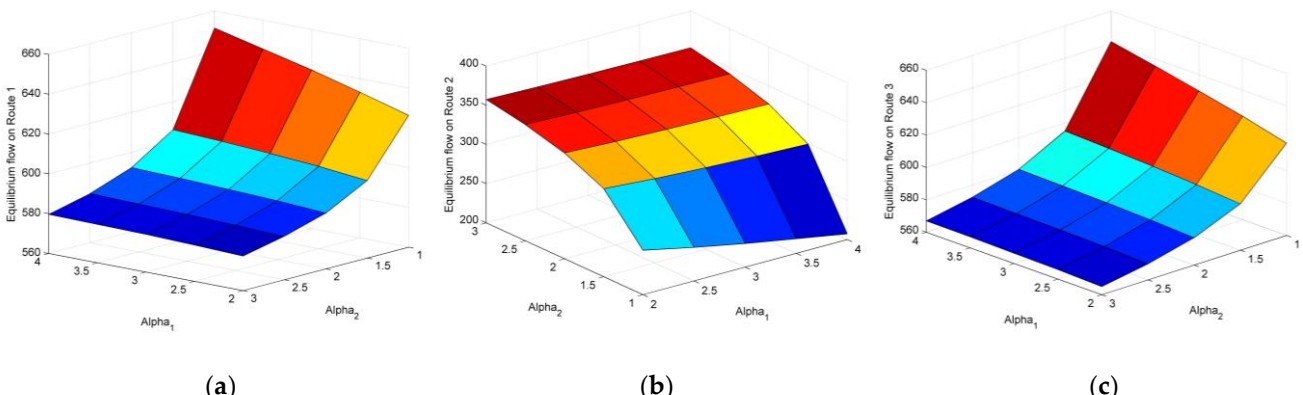

**Figure 6.** Testing results of equilibrium flow with regard to changing $\alpha_1$ and $\alpha_2$: (**a**) route 1; (**b**) route 2; and (**c**) route 3.

We further discuss the changes of TAPt and TAPl by changing the value of $\alpha_1$ and $\alpha_2$, which is shown in Figures 7a–c and 8a–c. From these figures, we see that as the value of $\alpha_1$ increases, the value of TAPt and TAPl on routes where the target set for travel cost can be reached successfully, will decrease, while the value of TAPt and TAPl on these routes always decreases with the value of $\alpha_2$ grows. We also see that on these routes, travelers will transform into less risk-averse and are very likely to arrive within the allowable delay of the company when $\alpha_2$ is larger, and travelers will become risk-seeking and bear the risk of violating the allowable delay when $\alpha_2$ is smaller. However, for some routes where the target set for travel cost fails to be reached, we see that TAPt is always 0.95, i.e., the probability that travelers will arrive on time, as well as the value of TAPl is almost 1. By properly increasing the value of $\beta_b$ and $\beta_s$, we can also run this testing in the situation that involves the perfect complementarity relationship, and obtain the similar trends as aforementioned.

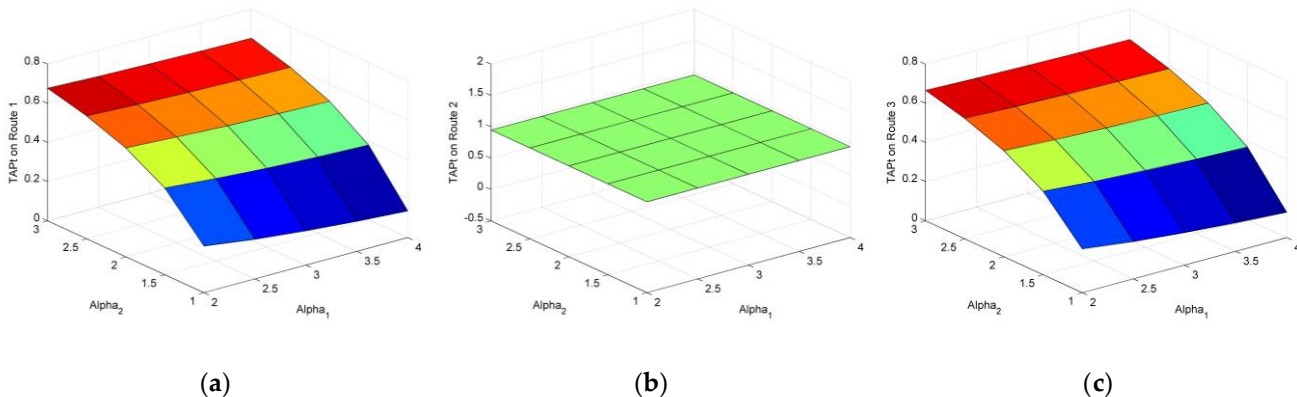

**Figure 7.** Testing results of TAPt with regard to changing $\alpha_1$ and $\alpha_2$: (**a**) route 1; (**b**) route 2; and (**c**) route 3.

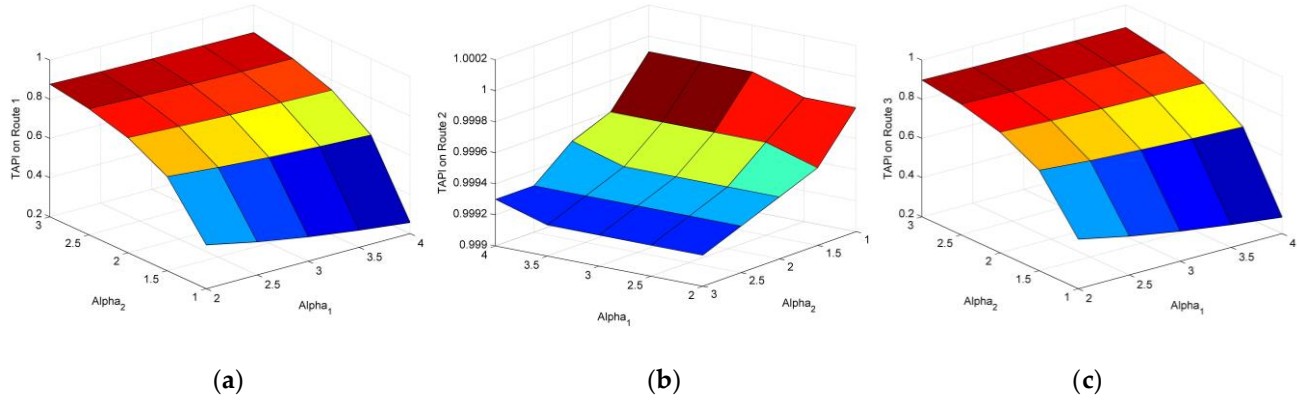

**Figure 8.** Testing results of TAPI with regard to changing $\alpha_1$ and $\alpha_2$: (**a**) route 1; (**b**) route 2; and (**c**) route 3.

### 4.1.5. Sensitivity Analysis via Changing $\beta_b$ and $\beta_s$

In this test, we show the impact of changing the value of complementarity ratios $\beta_b$ and $\beta_s$, i.e., we compare the imperfect complementarity relationship with the perfect complementarity relationship, and other parameter values follow the settings in the benchmark. The values of $\beta_b$ is changed from $1.5+\delta$ to 3, and the value of $\beta_s$ is changed from 1 to 2, where $\delta$ is a positive with small value. It can be verified that these settings of $\beta_b$ and $\beta_s$ satisfy the aforementioned relationship. The change of equilibrium route utility is shown in Figure 9, where its minimum value is 0.23, and its maximum value is 0.47. From this figure, and combining the equilibrium route utility shown in the above tests, we see that the value of equilibrium route utility on routes always decreases as the value of $\beta_b$ grows, and the value of equilibrium route utility on routes also decreases as the value of $\beta_s$ grows, but this decrease is smaller. Figure 9 shows the impact of different target interactions on the equilibrium route utility. For example, the interaction among three targets, i.e., changing the value of $\beta_b$, can bring fundamental changes to the equilibrium route utility, while the interaction between two targets, i.e., changing the value of $\beta_s$, can only bring relatively smaller changes to the equilibrium route utility.

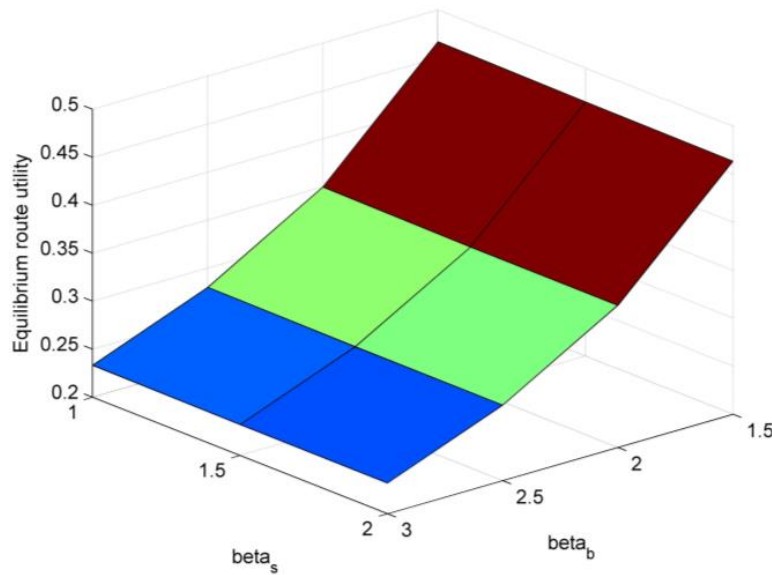

**Figure 9.** Testing results of equilibrium route utility with regard to changing $\beta_b$ and $\beta_s$.

The change of equilibrium route flows is presented in Figure 10a–c. From these figures, we can find that the increase in $\beta_b$ will result in flows transform from route 2 to route 1 and 3, as achieving the three targets simultaneously brings more utility to the travelers,

i.e., travelers are willing to realize these three targets at the same time, while the increase in $\beta_s$ will result in flows transform from route 1 and 3 to route 2, as interaction between two targets can weaken the impact of interaction among three targets, or weaken travelers' desire to achieve the three targets at the same time as shown in Equation (7).

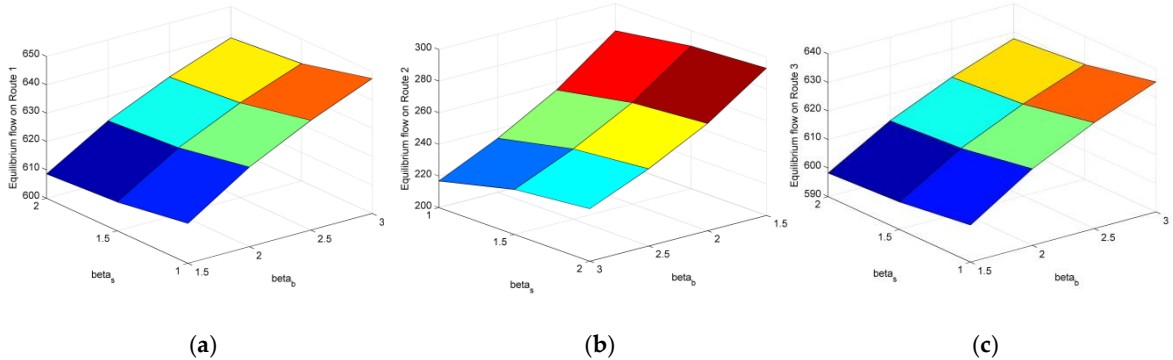

**Figure 10.** Testing results of equilibrium flow with regard to changing $\beta_b$ and $\beta_s$: (**a**) route 1; (**b**) route 2; and (**c**) route 3.

We further discuss the changes of TAPt and TAPl by changing the value of $\beta_b$ and $\beta_s$, which is shown in Figures 11a–c and 12a–c, respectively. From these figures, we see that the value of TAPt and TAPl on routes where the target set for travel cost can be reached successfully will decrease, while the value of $\beta_b$ grows, i.e., travelers tend to transform into more risk-seeking and bear the risk of violating the allowable delay of the company to achieve the three targets simultaneously, while the value of TAPt and TAPl on these routes always increases as the value of $\beta_s$ grows, i.e., interaction between two targets can weaken the impact of interaction among three targets. However, we see that TAPt or TAPl on the routes where the target set for travel cost fails to be reached is always 0.95, i.e., travelers' on-time arrival probability, and almost 1, respectively.

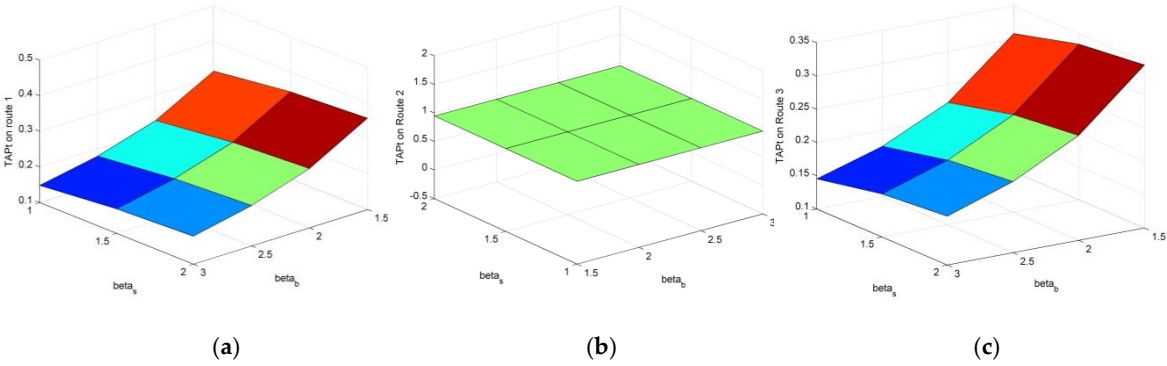

**Figure 11.** Testing results of TAPt with regard to changing $\beta_b$ and $\beta_s$: (**a**) route 1; (**b**) route 2; and (**c**) route 3.

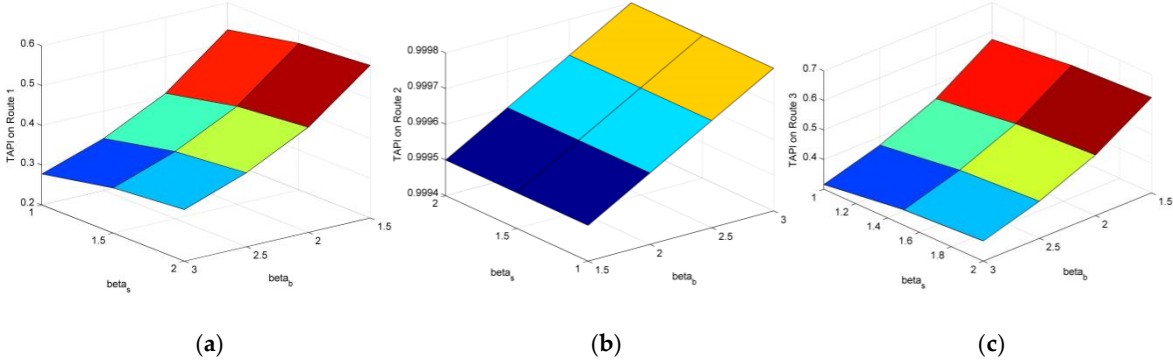

**Figure 12.** Testing results of TAPl with regard to changing $\beta_b$ and $\beta_s$: (**a**) route 1; (**b**) route 2; and (**c**) route 3.

### 4.2. Test on the Nguyen and Dupuis's Traffic Network

Finally, we apply our proposed model on a Nguyen and Dupuis's traffic network [48] shown in Figure 13, which can be seen as a more general network structure with several *OD* pairs. This traffic network consists of 13 nodes and 19 links. In this section, four *OD* pairs are taken into account, including $1 \rightarrow 2$, $1 \rightarrow 3$, $4 \rightarrow 2$, and $4 \rightarrow 3$, and the deterministic demands for all the *OD* pairs are all 800. Other characteristics of the network are shown in Table 2, where FFTT denotes free-flow travel time, and $\phi$ also reflects the link capacity degradation. Travelers' on-time arrival probability is 0.95, the target for route LAP is 5, and the target set for travel cost of route is 31.

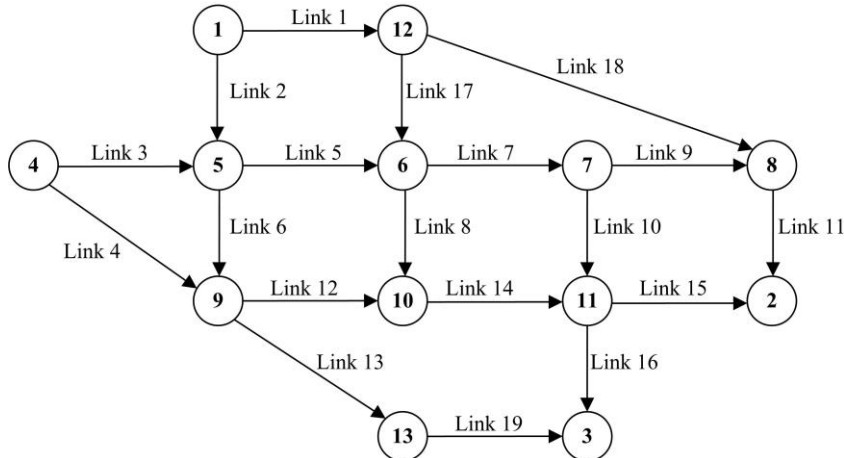

**Figure 13.** Nguyen and Dupuis's traffic network.

**Table 2.** Characteristics of the Nguyen and Dupuis's traffic network.

| Link Number | Free-Flow Travel Time | Capacity | Toll | $\phi$ |
|:---:|:---:|:---:|:---:|:---:|
| 1 | 5 | 600 | 3.5 | 0.8 |
| 2 | 12 | 400 | 8 | 0.9 |
| 3 | 7 | 400 | 5 | 0.8 |
| 4 | 10 | 400 | 6 | 0.8 |
| 5 | 8 | 600 | 6 | 0.7 |
| 6 | 6 | 500 | 4 | 0.8 |
| 7 | 10 | 500 | 7 | 0.6 |
| 8 | 10 | 400 | 8 | 0.8 |
| 9 | 8 | 600 | 5 | 0.8 |
| 10 | 6 | 500 | 4 | 0.9 |
| 11 | 12 | 400 | 9 | 0.9 |
| 12 | 11 | 600 | 8 | 0.9 |
| 13 | 6 | 600 | 4 | 0.7 |
| 14 | 8 | 500 | 5 | 0.8 |
| 15 | 9 | 500 | 6 | 0.8 |
| 16 | 10 | 400 | 6 | 0.9 |
| 17 | 12 | 400 | 10 | 0.8 |
| 18 | 8 | 600 | 5 | 0.9 |
| 19 | 10 | 500 | 7 | 0.7 |

We test two situations, namely in the imperfect complementarity relationship, where $\beta_b = \beta_s = 1$, and in the perfect complementarity relationship, where $\beta_b = 2$ and $\beta_s = 2$, $\alpha_1 = 3$, and $\alpha_2 = 2$. The convergence of our model is shown in Figure 14, where convergence of the first situation is shown in the left figure, and the convergence of the second situation is shown in the right figure. Specially, in order to better illustrate, we only display the first 500 iterations. As shown in these figures, we can find that the applicability and the performance of our proposed model, when it is embedded in a more general traffic

network. As aforementioned, similar testing results can be received by properly changing the values of the relevant parameters, and we omit them for brevity.

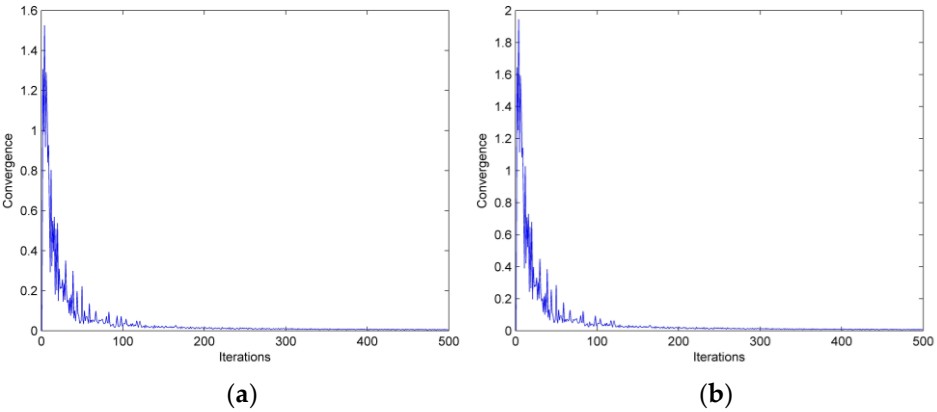

**Figure 14.** Convergence performance: (**a**) imperfect complementarity relationship; and (**b**) perfect complementarity relationship.

### 4.3. Insights of the Sensitivity Analysis

The main insights of the sensitivity analysis into route choice decision of travelers in the stochastic tolled network are summarized as follows.

(1) For some routes, where the target set for travel cost can be successfully reached (depending on travelers' acceptable expense), target achievement probability for travel time is smaller than travelers' on-time arrival probability, which means travelers become less risk-averse or even risk-seeking in order to achieve the three targets simultaneously. Meanwhile, on these kinds of routes, target achievement probability for late arrival penalty is also not very large, unless the allowable delay of the company is large; the relative significance between the achievements of the target set for travel cost and travel time is relatively large, or the extent of target interaction among three targets is small. That is, travelers usually bear the risk of violating the allowable delay of the company in order to achieve the three targets simultaneously, unless the aforementioned situations happen.

(2) For some routes, where the target set for travel cost fails to be reached (depending on travelers' acceptable expense), the probability of achieving the target for travel time is travelers' on-time arrival probability, and the probability of achieving the target for late arrival penalty is almost 1, which are always valid in all the testing.

(3) The equilibrium route flow distribution changes when the values of the three targets, the relative importance between different target achievements, and the target interaction change. Usually, travelers prefer the routes where the target set for travel cost can be successfully reached, which can be impacted by their risk-averse attitude, and the allowable delay of the company, but our testing shows that these impacts are not significant. If achieving the target for travel cost takes on greater importance, flows will shift from the routes where this target cannot be achieved to the routes where this target can be achieved in general, but the target achievements of travel time and late arrival penalty have impact on this flow shift. Meanwhile, the impact of the former one is larger. When the extent of the interaction among three targets become stronger, more travelers will choose the routes where the three targets can be achieved simultaneously, but stronger extent of the interaction between two targets could weaken this impact.

(4) The equilibrium route utility changes when the values of the three targets, the relative importance between different target achievements, and the target interaction change. From our testing, travelers' larger acceptable expense and stronger extent of risk-averse attitude leads to the larger equilibrium route utility, but the allowable delay of

the company has almost no impact on this. When the relative significance between the achievements of the target set for travel cost and travel time becomes larger, the equilibrium route utility is larger, but the target interaction, among three targets or between two targets, can decrease the equilibrium route utility.

## 5. Conclusions

In this paper, we studied the target-oriented multi-attribute route choice decision of travelers in the stochastic tolled traffic network, considering the impact of three attributes, which are (stochastic) travel time, (stochastic) late arrival penalty, and (deterministic) travel cost. In particular, we proposed a new route choice model, termed as a target-oriented multi-attribute travel utility model, for this analysis based on our recent methodology. In this new model, each attribute is assigned a target by travelers, and travelers' objective is to maximize their travel utility that is determined by the achieved targets. Moreover, the interaction between targets is interpreted as complementarity relationship between them, which can further affect their travel utility. Additionally, based on the travel utility model, target-oriented multi-attribute user equilibrium model is proposed, which is formulated as a variational inequality problem and solved with the method of successive average. Target for travel time is determined via travelers' on-time arrival probability, while targets for late arrival penalty and travel cost are given exogenously. Lastly, we apply the proposed model to the Braess and Nguyen–Dupuis traffic networks, and conduct sensitivity analysis of the parameters, including these three targets and the target inter-action between them.

There are several directions that merit the further study. (1) In this paper, tolls on the traffic network are given, and we will study the optimal congestion tolling on the basis of our proposed model at a later stage. (2) We will also adopt more behaviorally consistent methods to determine the value of the target set for late arrival penalty, travel cost, and travel time at a later stage.

**Author Contributions:** Conceptualization, X.Z.; methodology, X.Z.; formal analysis, X.Z., Z.G. and J.M.; writing—original draft preparation, X.Z.; writing—review and editing, Y.Z. and X.J.; funding acquisition, Y.Z. supervision, Y.Z. All authors have read and agreed to the published version of the manuscript.

**Funding:** This research was funded by National Natural Science Foundation of China (71974104).

**Institutional Review Board Statement:** Not applicable.

**Informed Consent Statement:** Not applicable.

**Data Availability Statement:** Not applicable.

**Acknowledgments:** Special thanks go to the anonymous reviewers for their suggestions which improve the quality of this paper.

**Conflicts of Interest:** The authors declare no conflict of interest.

## Appendix A

In the work of [40], link capacity is assumed to follow a uniform distribution. Upper bound of the capacity is the designed capacity $\bar{c}_a$ and lower bound of the capacity is the product of fraction $\phi_a$ and the designed capacity $\bar{c}_a$. Therefore, the distribution of link capacity is written as

$$C_a \sim U(\phi_a \cdot \bar{c}_a, \bar{c}_a) \tag{A1}$$

In Equation (A1), the link capacity degradation is reflected by $\phi_a$. The smaller the value of $\phi_a$ is, the less reliable the link is. If link capacity follows the uniform distribution constructed as above, route travel time will follow normal distribution [40], i.e.,

$$T_p^{rs} \sim N\left(E\left(T_p^{rs}\right), \sigma\left(T_p^{rs}\right)\right) \tag{A2}$$

where its mean and standard deviation are written as

$$E\left(T_p^{rs}\right) = \sum_a \left( \delta_{pa}^{rs} \cdot \left( t_a^0 + \beta t_a^0 x_a^n \frac{1 - \phi_a^{1-n}}{\bar{c}_a^n (1 - \phi_a)(1 - n)} \right) \right) \tag{A3}$$

and

$$\sigma\left(T_p^{rs}\right) = \sqrt{\sum_\alpha \left( \delta_{pa}^{rs} \cdot \beta^2 \left(t_a^0\right)^2 x_a^{2n} \left( \frac{1 - \phi_a^{1-2n}}{\bar{c}_a^{2n}(1 - \phi_a)(1 - 2n)} - \left( \frac{1 - \phi_a^{1-n}}{\bar{c}_a^n(1 - \phi_a)(1 - n)} \right)^2 \right) \right)} \tag{A4}$$

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
