# Peer review of "Target-Oriented User Equilibrium Considering Travel Time, Late Arrival Penalty, and Travel Cost on the Stochastic Tolled Traffic Network"

_sustainability, doi:10.3390/su13179992_

Round 1
Reviewer 1 Report
The paper covers the scope of the Journal. The Authors present a novel solution of an interesting and a contemporary problem, i.e. travelers’ multi-attribute route choice behavior. The Authors propose a methodology based on the target-oriented multi-attribute travel utility model and its application in two models, i.e. Braess traffic network composed of 4 nodes and 5 links, and Nguyen-Dupuis traffic network with 13 nodes and 19 links. It is worth to point out that not only the solution algorithm is an important aspect of the paper, but also the sensitivity analysis carried out in many dimensions. It provides a wide perspective of the analysis and the problem considered, as well. A conferred literature review is relevant.
The aspect of the implications for the policy design based on the obtained solutions is mentioned in the “Abstract”, although the Authors do not comment on that in the “Conclusions”.
The paper is well written. However, some minor grammar errors occur, e.g. the beginning of the sentence in a line 50 is written in lower case; there is „prblem” instead of „problem” in the line 347. A final revision of the text should be carried out.
Author Response
We are thankful to you for considering our paper and allowing us to revise our paper. We write a response report according to your comments, please see the attachment.

Reviewer 2 Report
The paper provides interesting insights on a novel, yet abstract equilibrium problem. Though I am supportive new traffic assignment methodologies such as this one, I feel there needs to be some clarification on the exact practical merits such that the “Target-oriented User Equilibrium”. I think with revision it will be a very good paper.
Specific Comments:
Abstract – This abstract is quite complicated containing a lot of technical jargon that isolates readership. I recommend this is re-written in simpler terms so that it can be accessible to a wider audience. Sentences like “The target for travel time is endogenously determined via travelers' exogenously given on-time arrival probability, while the targets for late arrival penalty and travel cost are exogenously given” – should be rephrased.
Introduction – Though the context is presented, there is a large body of literature that has been ignored and must be taken into consider. Works from Song Gao, Elise Miller-Hooks and S. Travis Waller concerning “adaptive stochastic traffic assignment methods” have been ignored and this work is linked and to some extent builds from such innovation. I recommend a new section be included to acknowledge the following papers at the very least.
- Gao, S., & Huang, H. (2012). Real-time traveler information for optimal adaptive routing in stochastic time-dependent networks. Transportation Research Part C: Emerging Technologies, 21(1), 196-213.
- Gao, S., Frejinger, E., & Ben-Akiva, M. (2010). Adaptive route choices in risky traffic networks: A prospect theory approach. Transportation research part C: emerging technologies, 18(5), 727-740.
- Unnikrishnan, A., & Waller, S. T. (2009). User equilibrium with recourse. Networks and Spatial Economics, 9(4), 575-593.
- Unnikrishnan, A., & Lin, D. Y. (2012). User equilibrium with recourse: continuous network design problem. Computer‐Aided Civil and Infrastructure Engineering, 27(7), 512-524.
- Boyles, S. D., & Waller, S. T. (2011). Optimal information location for adaptive routing. Networks and Spatial Economics, 11(2), 233-254.
- Miller‐Hooks, E. (2001). Adaptive least‐expected time paths in stochastic, time‐varying transportation and data networks. Networks: An International Journal, 37(1), 35-52.
- Wijayaratna, K. P., Dixit, V. V., Denant-Boemont, L., & Waller, S. T. (2017). An experimental study of the Online Information Paradox: Does en-route information improve road network performance?. Plos one, 12(9), e0184191.
- Wijayaratna, K. P., & Dixit, V. V. (2016). Impact of information on risk attitudes: Implications on valuation of reliability and information. Journal of choice modelling, 20, 16-34.
I recommend a new section be included to acknowledge the following papers at the very least.
Introduction (line 47 to 61) – This section is challenging to understand, it is unclear how the authors make the leap from travel time budgets and mean excess travel time to traffic assignment approaches. Travel time budgets are constraints/metrics used to define characteristics of users and it needs to be clarified how they are utilised in traffic assignment approaches.
Introduction (line 100 to 107) – It is not clear what the difference is between reference [28] and the research presented in this paper, is it the application of the method described in [28] or was there a modification of the methodology as well? This needs to be clarified.
Section 2 (line 147) – What are the implications of the assumption that all attributes are both stochastic and correlated? I feel this is a significant assumption in the modelling methodology without explaining the consequence clearly.
The formulation is quite sound from my perspective and this is the reason why I believe with some additions and revision of the paper it will be a valuable contribution to the literature. However, I am not quite sure about Equation (7) – the rewritten version of (3), is it an expansion?
The numerical analysis is interesting and the sensitivity analysis is very useful for micro-tolling/congestion pricing. However, the paper actually doesn’t clearly establish this until the conclusion. I think there would be a lot of value to include a sub-section at the end of the analysis which discusses practical applications of the novel assignment technique.
Overall, I believe this paper should be published after the above revisions are completed.
Author Response

(The authors gave the same response as above.)

Round 2
Reviewer 2 Report
The authors have addressed my concerns and I am happy with the revised paper. A proof reading of the final paper is recommended to address any grammatical errors.